

# Effects of direction specific exercise training on athletic performance: a systematic review and meta-analysis

Jiaru Huang[1], Tibor Hortobágyi[2], Thomas Dos'Santos[3], Yu Shi[1], Yilin Que[1], Junlei Lin[1], Yuying Su[1] and Wei Li[1]

[1] School of Strength and Conditioning Training, Beijing Sport University, Beijing, China
[2] Department of Kinesiology, Hungarian University of Sports Science, Budapest, Hungary
[3] Department of Sport and Exercise Sciences, The Manchester Metropolitan University, Manchester, United Kingdom

Corresponding author
Wei Li, liweiwin@sina.com

## ABSTRACT

**Background:** The similarity between movement patterns and force-vector specificity of training exercises and the target movement will likely result in the greatest transfer of the practiced skills and physical abilities to the intended sports skill performance. Therefore, this review aimed to investigate whether specific adaptations in athletic performance would be observed following direction specific exercise training.

**Methodology:** The literature search was performed in PubMed, Web of Science, and MEDLINE. Studies comparing acute (post-activation potentiation enhancement) and short-term (>2 weeks) effects of horizontally *vs.* vertically oriented resistance and plyometric training on athletic performance of recreationally active participants of either sex were included. The effect sizes were determined using a robust variance estimation random-effects model and were reported as Hedge's $g$.

**Results:** Twenty-two studies were included. For acute studies ($n = 4$), a small non-significant effect favoring horizontal training (HT) for sprint performance improvements ($g = -0.19$, $p = 0.17$) was evident. For short-term studies ($n = 18$), the results showed non-significant, small to large differences between HT and vertical training (VT) in pooled vertical and horizontal jump improvements ($g = 0.06$, $p = 0.67$), vertical ($g = 0.21$, $p = 0.17$) and horizontal jump ($g = -0.15$, $p = 0.40$), pooled vertical and horizontal maximal strength ($g = 0.27$, $p = 0.42$), horizontal ($g = -0.83$, $p = 0.16$) and vertical maximal strength ($g = 0.78$, $p = 0.28$), pooled short and medium distance sprint ($g = -0.23$, $p = 0.16$), short ($g = -0.33$ [$-0.85$, $0.19$], $p = 0.19$) and medium ($g = -0.12$ [$-0.37$, $0.13$], $p = 0.28$) distance sprint, and COD speed and maneuverability ($g = -0.45$, $p = 0.26$).

**Conclusions:** HT and VT were both equally effective in improving vertically and horizontally athletic performance, potentially refuting the theory of directional specificity of training on athletic performance outcomes.

# INTRODUCTION

From a mechanical perspective, the maximum transfer of training-induced adaptions to sports performance requires training routines that are specialized in terms of movement

patterns and force-vector specificity (*i.e.*, dynamic correspondence and coordinative overload) (*Hortobágyi et al., 2021*; *Worrell et al., 2001*). The similarity between movement patterns and force-vector specificity of training exercises and the target movement will likely result in the greatest transfer of the practiced skills and physical abilities to the intended sports skill performance (*Behm, 1995*; *Sale, 1988*). In addition to bioenergetic specificity (*Stone, Plisk & Collins, 2002*), training specificity also invokes kinematic and kinematic specificity (*i.e.*, mechanical specificity and coordinative overload) (*Stone, Plisk & Collins, 2002*). The theory assumes the similarity between the muscles that are activated during training and the muscles that are activated during the target movements for which the training is done to achieve training transfer (*Stone, Plisk & Collins, 2002*). This principle also extends to the type or regime of muscle work (positive, negative), rate and time of force generation, and the plane in which the training exercises are completed *vs.* the target movements are executed (*Verkhoshansky & Siff, 2009*); all of which are sub-components of dynamic correspondence. Recently, as part of the mechanical specificity of training, the directional specificity of force generation (force-vector specificity) has drawn some attention. Regarding the directional specificity of motion, there are two well-known theories, which are the force-vector (FV) theory and the dynamic correspondence (DC) theory (*Fitzpatrick, Cimadoro & Cleather, 2019*; *Goodwin & Cleather, 2016*).

The FV theory considers the direction of force application in the plane of global reference (*Zweifel, 2017*). The FV theory suggests that performing training exercises predominantly in the plane in which the target motor performance task is also executed is more effective compared with training during which training exercises and the target movements are done in different planes of movement (*Randell et al., 2010*). However, the DC theory states that forces exerted on the body are body-reference dependent (*i.e.*, local coordinate system) (*Goodwin & Cleather, 2016*). Both theories demonstrated direction-specific adaptions following exercise training: performing training exercises in the antero-posterior direction improves performance outcomes executed in the horizontal direction (*e.g.*, sprinting, and long jump), whilst performing training exercises in the vertical direction would preferentially improve performance outcomes executed vertically (vertical jump, squat). Therefore, training effects can be expected to transfer to performance outcomes more effectively if the training exercises and the target performance outcomes are done in the same directional orientation and less so if there is little to no correspondence between the direction of exercise done during training and in the target performance outcomes.

It is well documented that resistance training and plyometric training effectively improve neuromuscular and athletic performance (*Asadi et al., 2016*; *Fyfe, Hamilton & Daly, 2022*; *Lesinski, Prieske & Granacher, 2016*). Lower body plyometric exercises (activities which involve the stretch-shortening cycle such as repeated/rebound jumps) are typically power-dominant drills that can augment positive mechanical work generation through the stretch-shortening cycle (SSC) (*Taube et al., 2012*), typically improving reactive strength qualities. Resistance training (*i.e.*, training activities involving lifting an external load) can be used to improve a range of different force-velocity characteristics and

speed-dominant and force-dominant muscle qualities, depending on the relative load employed (*Suchomel et al., 2022*). Resistance training and weight training therefore serve distinct purposes, and in practice, both are often performed simultaneously in training sessions to improve overall muscular qualities. It is also documented that resistance training and plyometric training can improve neuromuscular and musculoskeletal function and these improvements can be 'transferred' to target performance outcomes such as sprinting and jumping regardless of the direction of the jump (*i.e.*, vertical or horizontal direction) (*Asadi et al., 2016*; *Fyfe, Hamilton & Daly, 2022*; *Lesinski, Prieske & Granacher, 2016*). Notably, both resistance training and plyometric exercises can be executed in various directions according to the direction of force application. These training exercises can be crudely categorized as those executed in the horizontal (HT) and vertical (VT) direction. For example, HT training exercises and target performance outcomes comprise horizontal jump (*i.e.*, standing long jump, serial horizontal jump) and hip thrust. In contrast, VT training comprises vertical jump and various forms of squatting variations.

Previous individual studies have reported inconsistent findings with respect to the directional specificity of training and subsequent transfer to athletic performance (*Gonzalo-Skok et al., 2019*). For example, HT was as effective as VT at enhancing performance outcomes in the vertical direction, yet HT was superior at increasing performance outcomes executed in the horizontal direction (*Gonzalo-Skok et al., 2019*). Moreover, study by *Nobari et al. (2023)* found that horizontal and vertical plyometric training were effective for improving change of direction (COD) and horizontal sprint performance, while greater improvements in COD and sprint time were observed following horizontal plyometric training, comparing to vertical plyometric training (*Dello Iacono et al., 2017*). Conflicting findings may be partially explained by individual differences in muscle qualities and movement technique. While elegantly reviewed previously (*Junge, Jørgensen & Nybo, 2023*; *Moran et al., 2021*), analyzing the directional effects of HT and VT on directionally categorized *vs.* directionally combined metrics of performance outcomes would increase our understanding of directional training specificity (*Junge, Jørgensen & Nybo, 2023*; *Moran et al., 2021*). The limitations of these reviews were that they did not analyze the acute intervention of HT and VT on subsequent performance outcomes, known as post-activation potentiation enhancement (PAPE), nor did they analyze the short-term intervention effects (>2 weeks) of HT and VT on maximal strength. Lastly, reviews used an inverse-variance random-effects meta-analysis (*Junge, Jørgensen & Nybo, 2023*), even though such a statistical model does not correct for the inclusion of multiple outcomes from one study (*Kadlec, Sainani & Nimphius, 2023*).

Taken together, the aim of this systematic review was to compare direction specific resistance and plyometric training (HT *vs.* VT) on athletic performance (*i.e.*, jump, sprint, maximal strength, and change of direction), vertically (*i.e.*, vertical jump), and horizontally oriented performance outcomes (*i.e.*, serial jumps) in recreationally active athletes. We hypothesized that both VT and HT will be equally beneficial at improving athletic performance tasks in both vertical and horizontally oriented outcomes. Critical review of existing research provides evidence-based recommendations for practitioners to optimize
training programs aimed at maximizing athletic performance and Such findings would tend to refute the directional specificity of training on performance outcomes.

# METHODS

The systematic review with meta-analysis was performed following the Preferred Reporting Items for Systematic Reviews and Meta-Analyses (PRISMA) guidelines (*Page et al., 2021*). A review protocol was not pre-registered for this review; however, the review methods were established prior to conducting the review.

## Search strategy

The literature search was conducted independently by JL and WL in PubMed, Web of Science, and MEDLINE (EBSCO) on September 06, 2023, with no date limits. The search items with Boolean search strategy were used: ("force application direction" OR "force-vector" OR "force application") OR ("vertical" OR "vertically" OR "horizontal" OR "horizontally") AND ("squat" OR "lunge" OR "leg press" OR "deadlift" OR "hip thrust") OR ("resistance training" OR "plyometric" OR "jump"). The studies included in two previous reviews were also screened by the full text to identify the eligibility for inclusion in the review (*Junge, Jørgensen & Nybo, 2023*; *Moran et al., 2021*). Only original peer-reviewed journal articles written in English were considered in the review.

## Inclusion and exclusion criteria

For eligibility in the review, the PICO strategy (population, intervention, comparison, and outcome) was used (*Morris et al., 2022*). Studies were included if they met the following criteria: (1) participants were recreationally active players of either sex; (2) interventions were performed in terms of resistance training or/and plyometric training, the duration of short-term intervention studies were ≥2 weeks; (3) studies compared acute or/and short-term effects of VT and HT; (4) studies reported pre-to-post changes (mean and SD) of at least one of the following athletic performance tasks: sprint completion time/speed, jump distance/height, maximal strength (1-repetition maximal, isometric strength), and change of direction (COD) and maneuverability completion time. Studies were excluded if they met any of the following criteria: (1) intervention involved a combination of vertical and horizontal exercise training; (2) the pre- and post-data were unavailable, and (3) were not a peer reviewed full published article in English using human participants.

## Study selection and data extraction

All search results were imported into reference management software (Endnote X9.3.2; Stanford University, Stanford, CA, USA). Duplicate records were removed using the automatic function of Endnote. Based on the inclusion and exclusion criteria, the titles and abstracts of the remaining studies were initially screened, and then the full text was assessed. The evaluation process was conducted independently and separately by two co-authors (JL and WL). In the event of a disagreement, a third assessor (TD) was approached. Data of included studies were extracted by one assessor (JL) and confirmed by another assessor (WL) into Microsoft Excel. The following information was extracted: general study information (author(s), year); characteristics of participants (sample size,

training status, sex, age, height, mass); intervention protocol (frequency and duration of intervention, exercise modality, intensity, and volume); testing protocols.

## Methodological quality

According to previous studies, a modified version of the Downs and Black Quality Index tool was used to evaluate study quality (*Downs & Black, 1998*; *Fox et al., 2018*). This tool has been used frequently in the sport science studies (*Emery et al., 2015*; *Johnston et al., 2018*) and allows to assess the methodological quality not only of randomized controlled trials but also non-randomized studies (*Downs & Black, 1998*; *Fox et al., 2018*). This tool has a high test-retest reliability (r = 0.88), internal consistency (k = 0.89), and good inter-rater reliability (r = 0.75) (*Downs & Black, 1998*). This assessment tool consists of three assessment components and 10 items, including reporting, external validity and internal validity bias. Two independent co-authors (YS and YL) assessed 10 items of the checklist, each with a score of one (clearly yes) or no point (clearly no or unable to determine). Arbitration of dissenting opinions by a third author (WL). A total score was 10 points, with a higher score representing the higher methodological quality of the study.

## Statistical analysis

All analyses were performed using robumeta (version 2.1) and metafor (version 4.2.0) in R version 4.2.3 (*R Core Team, 2023*). Data from a minimum of three outcomes from the identified studies were reported in the meta-analysis. Within-group effect size (ES) was calculated to quantify the improvement magnitude following HT and VT, and between-group ES was used to compare the differences in the improvement magnitude between HT and VT. For all analyses, we used a robust variance estimation (RVE) random-effects meta-analysis method. Effect sizes were presented as Hedge's *g* with 95% confidence intervals (CI), with effect sizes 0.15–0.39, 0.40–0.74, and ≥0.75 interpreted as small, moderate and large, respectively (*Brydges, 2019*). Subgroup analyses were conducted to determine the potential influence of moderator variables, which included intervention duration (≥6 weeks *vs.* <6 weeks), age (≥18 years *vs.* <18 years), and total sessions (≥12 times *vs.* <12 times), and intervention modalities (resistance training *vs.* plyometric training). Moreover, single-factor analysis for sprint distance, jump, and maximal strength tests were undertaken. The classification of sprint distance was determined by previous studies (short distance: ≤20 m, medium distance: >20 and ≤40 m) (*Barnes et al., 2014*; *Johnston, Gabbett & Jenkins, 2014*), and the jump and maximal strength tests were divided into horizontal and vertical tasks. The effect sizes of sprint and COD speed were standardized, so the positive ES represented improvements in sprint and COD performance.

Heterogeneity was evaluated using Tau-squared ($Tau^2$) and the I-squared test ($I^2$). $I^2$ values were used to assess statistical heterogeneity, and the values of 25%, 50%, and 75% were identified as small, moderate, and high heterogeneity, respectively (*Higgins et al., 2003*). To measure the asymmetry of the funnel plot and to evaluate the risk of publication bias, Egger's test of the intercept was used (*Egger et al., 1997*). Qualitative analysis of funnel plots was performed when the sample size was ≥ ten (*Sterne et al., 2011*). An asymmetrical

funnel plot and a statistically significant Egger's test ($p \leq 0.05$) demonstrate the occurrence of small study bias.

## RESULTS

### Search results

A total of 5,616 potentially relevant studies were identified, of which 510 were removed due to duplication. After screening titles and abstracts, 5,081 studies were excluded. Following evaluation of the full text of the remaining studies, 20 studies met the pre-determined inclusion criteria. Moreover, two additional studies were included through reference lists searching of the previous reviews (*Junge, Jørgensen & Nybo, 2023*; *Moran et al., 2021*). Finally, 22 studies were included in this meta-analysis, as presented in Fig. 1.

### Methodological quality

The quality scores of the studies ranged from 7 to 9, with an average of 8.6. All studies included in this review were categorized as high methodological quality; thus, none were removed due to low quality (Table 1).

### Study characteristics

The overall number of participants included in these studies was 578, of whom 61 were female and 517 were male. For individual studies, the sample sizes ranged from 14 to 45 participants, with an average of 26.3. Details were displayed in Table 2.

Four studies explored the acute effects of HT and VT on athletic performance outcomes (*Atalağ et al., 2020*; *Carbone et al., 2020*; *Dello Iacono, Martone & Padulo, 2016*; *Fernández-Galván et al., 2022*). For these four studies, the average age of all participants was 19.7 year. Only one study included both female and male (*Atalağ et al., 2020*), while the remaining studies analyzed only male (*Carbone et al., 2020*; *Dello Iacono, Martone & Padulo, 2016*; *Fernández-Galván et al., 2022*). Plyometric training was conducted in one study (*Dello Iacono, Martone & Padulo, 2016*), and the intervention load involved three sets × five repetitions vertical or horizontal single-leg 25 cm drop jump. Resistance training was performed in terms of squat *vs.* hip thrust in three studies (*Atalağ et al., 2020*; *Carbone et al., 2020*; *Fernández-Galván et al., 2022*), the intervention loads of 1–3 sets × 3 repetitions × 85–90% 1 repetition maximal (RM) were performed.

Eighteen studies investigated the short-term effects of HT *vs.* VT with a total of 507 participants (455 males and 52 females) (*Abade et al., 2021*; *Asencio et al., 2022*; *Aztarain-Cardiel et al., 2023*; *Barbalho et al., 2020*; *Contreras et al., 2017*; *Dello Iacono et al., 2017*; *Gonzalo-Skok et al., 2019*; *Hammond et al., 2019*; *Keller et al., 2020*; *Kurt et al., 2023*; *Los Arcos et al., 2014*; *Loturco et al., 2015*; *Manouras et al., 2016*; *Nobari et al., 2023*; *Ramírez-Campillo et al., 2015*; *Talukdar et al., 2022*; *Tibor, Judit & Zoltán, 1990*; *Wilson et al., 2022*). Maximal strength (horizontal: three studies; vertical: four studies), jump (horizontal: eight studies; vertical: 12 studies), sprint, and COD and maneuverability performance were reported in four (*Asencio et al., 2022*; *Barbalho et al., 2020*; *Contreras et al., 2017*; *Hammond et al., 2019*; *Talukdar et al., 2022*), 12 (*Abade et al., 2021*; *Asencio et al., 2022*; *Aztarain-Cardiel et al., 2023*; *Contreras et al., 2017*; *Dello Iacono et al., 2017*; *Gonzalo-Skok et al., 2019*;

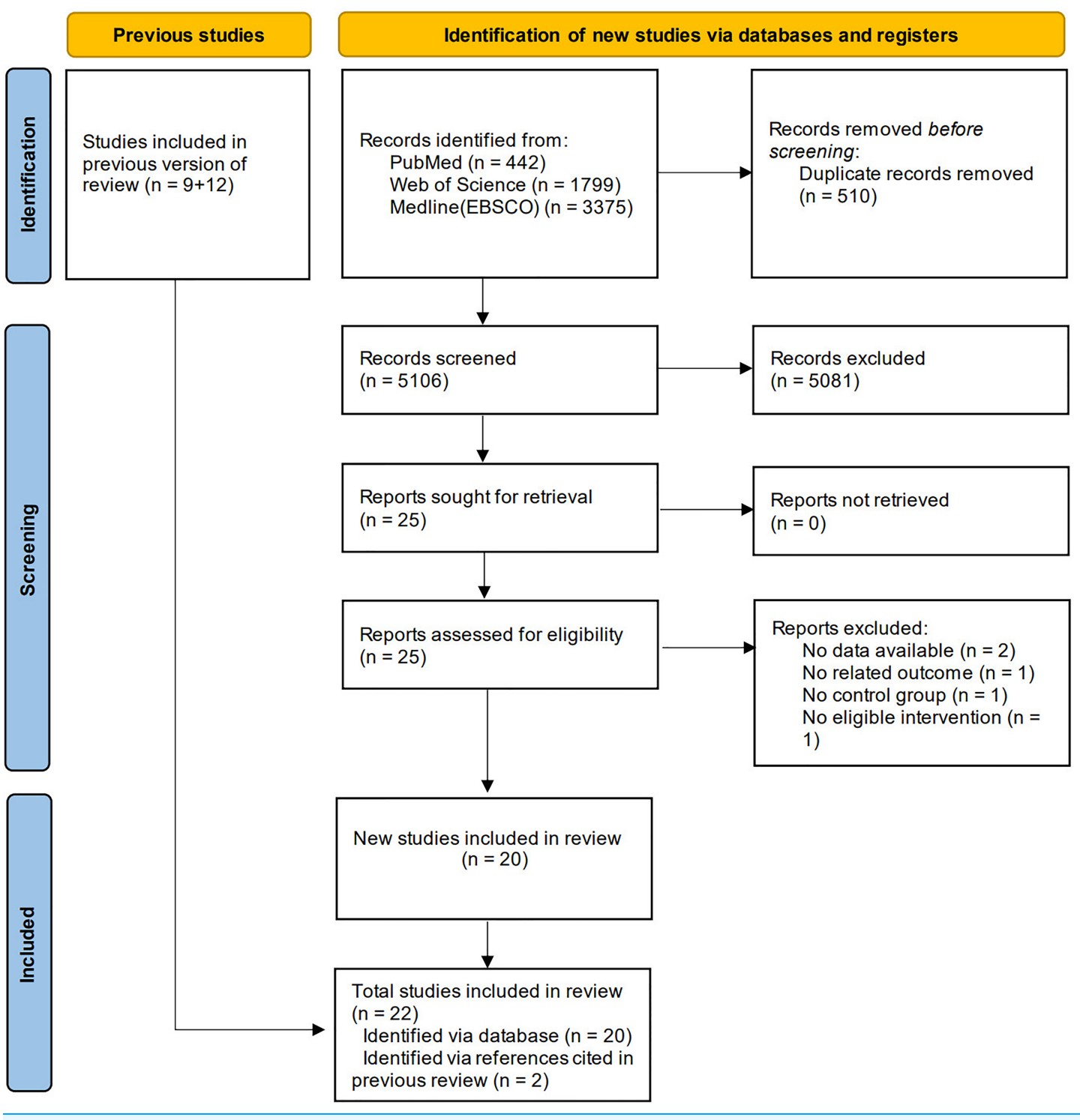

**Figure 1 Flow chart for inclusion and exclusion of studies.**

*Keller et al., 2020*; *Kurt et al., 2023*; *Los Arcos et al., 2014*; *Loturco et al., 2015*; *Manouras et al., 2016*; *Ramírez-Campillo et al., 2015*; *Talukdar et al., 2022*; *Tibor, Judit & Zoltán, 1990*; *Wilson et al., 2022*), nine (*Abade et al., 2021*; *Aztarain-Cardiel et al., 2023*; *Contreras et al.,*

**Table 1 Results of methodological quality of the included studies.**

| Item | 1 | 2 | 3 | 4 | 5 | 6 | 7 | 8 | 9 | 10 | Score |
|---|---|---|---|---|---|---|---|---|---|---|---|
| **Acute studies** | | | | | | | | | | | |
| *Atalağ et al. (2020)* | 1 | 1 | 1 | 1 | 1 | 1 | ? | 1 | 1 | 1 | 9 |
| *Carbone et al. (2020)* | 1 | 1 | 1 | 1 | 1 | 0 | ? | 1 | 1 | 1 | 8 |
| *Dello Iacono, Martone & Padulo (2016)* | 1 | 1 | 1 | 1 | 1 | 1 | ? | 1 | 1 | 1 | 9 |
| *Fernández-Galván et al. (2022)* | 1 | 1 | 1 | 1 | 1 | 1 | ? | 1 | 1 | 1 | 9 |
| **Short-term studies** | | | | | | | | | | | |
| *Abade et al. (2021)* | 1 | 1 | 1 | 1 | 1 | 1 | ? | 1 | 1 | 1 | 9 |
| *Asencio et al. (2022)* | 1 | 1 | 1 | 1 | 0 | 1 | ? | 1 | 1 | 1 | 8 |
| *Aztarain-Cardiel et al. (2023)* | 1 | 1 | 1 | 1 | 1 | 1 | ? | 1 | 1 | 1 | 9 |
| *Barbalho et al. (2020)* | 1 | 1 | 1 | 1 | 1 | 1 | ? | 1 | 1 | 1 | 9 |
| *Contreras et al. (2017)* | 1 | 1 | 1 | 1 | 0 | 1 | ? | 1 | 1 | 1 | 8 |
| *Dello Iacono et al. (2017)* | 1 | 1 | 1 | 1 | 1 | 1 | ? | 1 | 1 | 1 | 9 |
| *Gonzalo-Skok et al. (2019)* | 1 | 1 | 1 | 1 | 1 | 1 | ? | 1 | 1 | 1 | 9 |
| *Hammond et al. (2019)* | 1 | 1 | 1 | 1 | 1 | 1 | ? | 1 | 1 | 1 | 9 |
| *Keller et al. (2020)* | 1 | 1 | 1 | 1 | 0 | 1 | ? | 1 | 1 | 1 | 8 |
| *Kurt et al. (2023)* | 1 | 1 | 1 | 1 | 1 | 1 | ? | 1 | 1 | 1 | 8 |
| *Los Arcos et al. (2014)* | 1 | 1 | 1 | 1 | 1 | 1 | ? | 1 | 1 | 1 | 9 |
| *Loturco et al. (2015)* | 1 | 1 | 1 | 1 | 1 | 1 | ? | 1 | 1 | 1 | 9 |
| *Manouras et al. (2016)* | 1 | 1 | 1 | 1 | 1 | 1 | ? | 1 | 1 | 1 | 9 |
| *Nobari et al. (2023)* | 1 | 1 | 1 | 1 | 0 | 1 | ? | 1 | 1 | 1 | 8 |
| *Ramírez-Campillo et al. (2015)* | 1 | 1 | 1 | 1 | 1 | 1 | ? | 1 | 1 | 1 | 9 |
| *Talukdar et al. (2022)* | 1 | 1 | 1 | 1 | 1 | 0 | ? | 1 | 1 | 1 | 8 |
| *Tibor, Judit & Zoltán (1990)* | 1 | 1 | 1 | 1 | 0 | 0 | ? | 1 | 1 | 1 | 7 |
| *Wilson et al. (2022)* | 1 | 1 | 1 | 1 | 1 | 1 | ? | 1 | 1 | 1 | 9 |
| Average score = 8.59, Median score = 8 | | | | | | | | | | | |

**Note:**
Item: 1. the objectives of the study were clearly reported, 2. the main outcomes to be assessed were clearly reported, 3. the characteristics of the participants were clearly reported, 4. the main ûndings were clearly reported, 5. the estimates of the random variability in the data for the main outcomes were clearly reported, 6. the actual probability values were clearly reported, 7. can the participants represent the entire population, 8. if any of the results of the study were based on 'data dredging' was this made clear?, 9. were the statistical tests appropriate, 10. were the main outcome measure accurate. 1 the item was clearly reported, 0 the item was not clearly reported, ? unknown.

*2017*; *Dello Iacono et al., 2017*; *Gonzalo-Skok et al., 2019*; *Kurt et al., 2023*; *Los Arcos et al., 2014*; *Loturco et al., 2015*; *Manouras et al., 2016*; *Nobari et al., 2023*; *Ramírez-Campillo et al., 2015*; *Talukdar et al., 2022*) and eight studies (*Asencio et al., 2022*; *Aztarain-Cardiel et al., 2023*; *Dello Iacono et al., 2017*; *Gonzalo-Skok et al., 2019*; *Keller et al., 2020*; *Kurt et al., 2023*; *Nobari et al., 2023*; *Ramírez-Campillo et al., 2015*), respectively. Six studies involved resistance training (*Abade et al., 2021*; *Asencio et al., 2022*; *Barbalho et al., 2020*; *Contreras et al., 2017*; *Hammond et al., 2019*; *Wilson et al., 2022*), eleven studies used plyometric training (*Aztarain-Cardiel et al., 2023*; *Dello Iacono et al., 2017*; *Gonzalo-Skok et al., 2019*; *Keller et al., 2020*; *Kurt et al., 2023*; *Loturco et al., 2015*; *Manouras et al., 2016*; *Nobari et al., 2023*; *Ramírez-Campillo et al., 2015*; *Talukdar et al., 2022*; *Tibor, Judit & Zoltán, 1990*), and one study analyzed combined resistance and plyometric training (*Los Arcos et al., 2014*).

**Table 2 Details of studies included in this review.**

| Study | Participants | Vertical intervention | Horizontal intervention | Set/repetition/intensity | Duration/frequency | Main tests |
|---|---|---|---|---|---|---|
| Dello Iacono, Martone & Padulo (2016) | N = 18, male; 19.6 ± 0.5 yr; 182.5 ± 6.4 cm; 83.8 ± 8.4 kg; U-20 handball national team | Vertical one-leg DJ | Horizontal one-leg DJ | 3s × 5r | PAP intervention (8 min) | CMJ, 25 m shuttle sprint (12.5 + 12.5 and 180° COD) |
| Carbone et al. (2020) | N = 17, male; 22.14 ± 2.52 yr; 81.06 ± 9.6 kg; 1.78 ± 0.05 m; Amateur rugby players | Back squat | Hip thrust | 3s × 3RM | PAP intervention (8 min) | 5 m sprint, 10 m sprint |
| Ataláğ et al. (2020) | N = 17; male: N = 8; female: N = 9; 21.71 ± 1.49 yr; 173.97 ± 11.88 cm; 75.58 ± 11.89 kg; kinesiology and exercise science students | Back squat | Hip thrust | 1s × 3r × 90% 1RM | PAP intervention (8 min) | Vertical jump, 20 yard dash, 40 yard dash |
| Fernández-Galván et al. (2022) | N = 19, male; 15.61 ± 1.35 yr; 173.89 ± 8.24 cm; 68.31 ± 13.34 kg; tennis players | Full squat | Hip thrust | 1s × 3r × 85% 1RM | PAP intervention (4 min) | 5 m sprint; 10 m sprint; 30 m sprint |
| Wilson et al. (2022) | N = 33, male; BS: N = 11; 179.3 ± 6.0 cm; 79.0 ± 17.9 kg; DL: N = 11; 180.2 ± 6.7 cm; 78.3 ± 7.0 kg; HT: N = 11; 182.8 ± 5.8 cm; 81.0 ± 11.5 kg; recreational sports participants | Main lift: back squat or deadlift | Main lift: hip thrust | 3–5s × 4–10r × 70–85% 1RM | 6 weeks, 2 times/week | Horizontal jump, vertical jump |
| Ramírez-Campillo et al. (2015) | N = 40, male; VG: N = 10, 11.6 ± 1.4 yr, 144 ± 9.6 cm, 40 ± 5.9 kg, HG: N = 10, 11.4 ± 1.9 ys, 150 ± 12.3 cm, 44.6 ± 11.0 kg | CMJ | Horizontal CMJ | 3–6s × 5–10r | 6 weeks, 2 times/week | CMJ, HCMJ; 15 m sprint, 30 m sprint, CODs |
| Kurt et al. (2023) | N = 32, male, VG: N = 11, HG: N = 11, CG: N = 10, 12.09 ± 0.89 yr, U11–U13 leagues soccer players | Vertical ankle jump, front obstacle jump, CMJ, tuck jump, foot fire, pogo tuck jump, squat jump | Horizontal obstacle training, heidens, broad jump, kneeling jump, kneeling to broad jump, diagonal jump, multiple diagonal jump | 4–6s × 4–5r | 6 weeks, 2 times/week | Vertical jump, SLJ, 10 m sprint, 20 m sprint, Pro-agility |
| Contreras et al. (2017) | N = 28, male, HG: N = 14, 15.49 ± 1.16 yr, 178.73 ± 5.02 cm, 78.32 ± 12.47 kg, VG: N = 14, 15.48 ± 0.74 yr, 181.61 ± 5.51 cm, 81.16 ± 12.37 kg, rugby and rowing athletes | Front squat | Hip thrust | 4s × 6–12r × 6-12RM | 6 weeks, 2 times/week | Vertical jump, horizontal jump, 10 m sprint, 20 m sprint, hip thrust, front squat, IMTP |
| Abade et al. (2021) | N = 24, male, 16.56 ± 0.56 yr, 176.3 ± 5.8 cm, 66.6 ± 6.2 kg, HG: N = 8, VG: N = 8, CG: N = 8, U17 football players | Back half squat | Hip thrust | 3s × 4-10RM | 20 weeks, 1 time/week | Squat jump, CMJ, horizontal jump, 10m sprint, 20 m sprint |

(Continued)

| Study | Participants | Vertical intervention | Horizontal intervention | Set/repetition/intensity | Duration/frequency | Main tests |
|---|---|---|---|---|---|---|
| Asencio et al. (2022) | N = 29, male, VG: N = 11, 22.6 ± 4.0 yr, 1.79 ± 0.09 m, 75.5 ± 14.1 kg, HG: N = 8, 22.0 ± 4.1 yr, 1.76 ± 0.05 m, 74.9 ± 11.4 kg, 1.77 ± 0.09 m, 75.6 ± 16.8 kg recreational athletes | Vertical flywheel training | Horizontal flywheel training | 3s × 8r × 0.045/0.090 kg·m$^2$ | 4 weeks, 2 times/week | 1RM squat, CMJ, 505-dominant side test, 505-non-dominant side test |
| Gonzalo-Skok et al. (2019) | N = 20, male, 13.2 ± 0.7 yr, 59.5 ± 12.7 kg, 172.9 ± 7.9 cm, VG: N = 9, HG: N = 9, highly trained basketball players | 20 cm DJ, squat jump, CMJ, tuck jump, hurdle jump | Horizontal DJ 10 cm, standing long jump, unilateral jump, triple jump | 3–5s × 2–5r | 6 weeks, 2 times/week | 5, 10, 25 m sprint, bilateral and unilateral CMJ, horizontal unilateral CMJ, V-cut, 180° COD |
| Hammond et al. (2019) | N = 14, male, VG: N = 7, HG: N = 7, 22.07 ± 0.62 yr, 179.31 ± 6.96 cm, 79.77 ± 13.81 kg, trained subjects | Back squat | Hip thrust | 3s × 80% 1RM × perform to failure | 4 weeks, 2 times/week | 1RM squat, 1RM hip thrust |
| Manouras et al. (2016) | N = 30, male; VG: N = 10, 20.75 ± 6.14 yr, 1.78 ± 0.07 m, 72.10 ± 11.30 kg, HG: N = 10, 19.10 ± 5.75 yr, 1.75 ± 0.06 m, 69.80 ± 11.00 kg, soccer players | Vertical ankle jump, CMJ, front obstacle jump, DJ | Horizontal ankle jump, long jump, diagonal obstacle jump, multiple long jump | 3–5s × 4–6r | 8 weeks, 1 time/week | 10, 30 m sprint, SLJ, CMJ |
| Loturco et al. (2015) | N = 24, male, VG: N =12, 18.2 ± 0.6 yr, 177 ± 5.2 cm, 72.7 ± 5.6 kg, HG: N = 12, 18.5 ± 0.8 yr, 176 ± 4.3 cm, 71.8 ± 4.2 kg, U-20 soccer players | CMJ | Horizontal jump | 4–6s × 8–10r | 3 weeks, 6 times/week | CMJ, horizontal jump, 10, 20 m sprint |
| Dello Iacono et al. (2017) | N = 18, male, 23.4 ± 4.6 yr, 192.5 ± 3.7 cm, 87.8 ± 7.4 kg, well-trained handball players | Vertical DJ | Horizontal DJ | 5–8s × 6–10r | 10 weeks, 2 times/week | CMJ, 10, 25 m sprint, COD |
| Barbalho et al. (2020) | N = 22, female, VG: N = 12, 26.4 ± 1.3 yr, 171.8 ± 3.8 cm, 69.5 ± 4.9 kg, HG: N = 10, 27.5 ± 1.4 yr, 170.8 ± 4.4 cm, 67.5 ± 4.7 kg, well-trained subjects | Back squat | Hip thrust | 6s × 4–15r | 12 weeks, 1 time/week | 1RM squat, 1RM hip thrust |
| Talukdar et al. (2022) | N = 30, female, VG: N = 11, 13.50 ± 0.96 yr, 56.25 ± 14.87 kg, 1.64 ± 0.10 m, HG: N=10, 13.40 ± 0.92 yr, 54.73 ± 7.16 kg, 1.65 ± 0.06 m, student athletes and physical education students | Vertical plyometric training | Horizontal plyometric training | 3–5s × 5–10r | 7 weeks, 2 times/week | 10, 20, 30 m sprint, IMTP, vertical jump, SLJ |

| Study | Participants | Vertical intervention | Horizontal intervention | Set/repetition/intensity | Duration/frequency | Main tests |
|---|---|---|---|---|---|---|
| *Los Arcos et al. (2014)* | $N = 15$, male, VG: $N = 7$, 20.3 ± 1.9 yr, 1.8 ± 0.1 m, 69.6 ± 4.1, HG: $N = 8$, 19.6 ± 1.6 yr, 1.8 ± 0.1 m, 71.2 ± 5.4 kg, professional soccer players | Vertical half squat, vertical calf exercises, vertical jump, | Horizontal half squat, horizontal calf exercises, sled walking, horizontal jump, sled towing | Resistance training: 2s × 5r × 30–76% body weight plyometric training: 1s × 5r × 0–5% body weight | 8 weeks 1–2 times/week | CMJ, CMJ-single leg, 5, 15 m sprint |
| *Keller et al. (2020)* | $N = 45$, male, 14 ± 0.8 yr, 63 ± 14 kg, 175 ± 11 cm, HG: $N = 12$, VG: $N = 12$, U15 team sport athletes | DJ, drop landing, split-squat jump, multiple bounding | Horizontal jump, lateral jump | HG: 4–7s × 5r, VG: 3s × 8–10r × 70–100 cm | 4 weeks, 2 time/week | CMJ, DJ, SLJ |
| *Tibor, Judit & Zoltán (1990)* | $N = 40$, male, 13.4 ± 0.11 yr, VG: $N = 15$, HG: $N = 15$, Control G: $N = 10$, healthy and active boys | Vertical plyometric training | Horizontal plyometric training | 80–180 jump/jump/session, 160–360 jumps/week | 10 weeks, 2 times/week | SLJ, 5 horizontal jumps, CMJ |
| *Nobari et al. (2023)* | $N = 23$, male, 23.1 ± 2.8 yr, 178.3 ± 4.8 cm, 72.4 ± 4.9 kg, VG: $N = 10$, HG: $N = 9$, semi-professional soccer players | Vertical ankle jump, CMJ, squat jump | Horizontal ankle jump, long jump, diagonal obstacle jump, | 4s × 6–7r | 13 weeks, 5–6 times/week | 10, 20, 30 m sprint, 5-0-5 test |
| *Aztarain-Cardiel et al. (2023)* | $N = 40$, male, VG: $N = 11$, 21.2 ± 2.0 yr, 1.90 ± 0.08 m, 87.1 ± 17.7 kg, HG: $N = 10$, 23.1 ± 3.7 yr, 1.89 ± 0.08 m, 85.3 ± 12.9 kg, basketball players | Vertical plyometric | Horizontal plyometric | 2s × 3–12r | 6 weeks, 2 times/week | Rocket jump, abalakov jump, horizontal jump, 5, 10, 20 m sprint |

**Note:**
VG, vertical group; HG, horizontal group; DJ, drop jump; SLJ, standing long jump; CMJ, countermovement jump; HCMJ, horizontal countermovement jump; s, sets; r, repetitions; RM, repetition maximal; IMTP, isometric mid-thigh pull; COD, change of direction.

The average interventions length of interventions was 7.7-weeks, ranging from 3 to 20 weeks. The interventions were conducted twice a week in 12 studies (*Asencio et al., 2022*; *Aztarain-Cardiel et al., 2023*; *Contreras et al., 2017*; *Dello Iacono et al., 2017*; *Gonzalo-Skok et al., 2019*; *Hammond et al., 2019*; *Keller et al., 2020*; *Kurt et al., 2023*; *Ramírez-Campillo et al., 2015*; *Talukdar et al., 2022*; *Tibor, Judit & Zoltán, 1990*; *Wilson et al., 2022*), and 5–6 times per week in two studies (*Loturco et al., 2015*; *Nobari et al., 2023*), respectively, while 1 and 1–2 times per week in 3 (*Abade et al., 2021*; *Barbalho et al., 2020*; *Manouras et al., 2016*) and one studies (*Los Arcos et al., 2014*), respectively.

## Meta-analyses results

For within-group meta-analyses, small and non-statistically significant changes in pooled sprint performance across the various distances were observed following acute VT ($g = -0.17$ [$-0.59, 0.25$], $p = 0.33$) and acute HT ($g = -0.31$ [$-0.83, 0.22$], $p = 0.18$). Small statistical heterogeneity was observed ($I^2 = 0$–$43\%$, $Tau^2 = 0$–$0.1$). For the between-group meta-analysis, there was a small and non-significant effect in favor of acute HT for improving sprint performance ($g = -0.19$ [$-0.51, 0.13$], $p = 0.17$) (Fig. 2). Small statistical heterogeneity was observed in sprint comparison ($I^2 = 0\%$, $Tau^2 = 0$).

For pooled vertical and horizontal jump performance, moderate and significant improvements were observed following VT ($g = 0.50$ [$0.19, 0.81$], $p = 0.003$) and HT ($g = 0.62$ [$0.33, 0.91$], $p = 0.0004$). Moderate to high statistical heterogeneity was found (VT: $I^2 = 49.25$, $Tau^2 = 0.20$; HT: $I^2 = 78.41$, $Tau^2 = 0.81$). For the between-group meta-analysis, a small and non-significant effect favoring VT ($g = 0.06$ [$-0.22, 0.33$], $p = 0.67$) was evident (Fig. 3). Small statistical heterogeneity was found ($I^2 = 46.46$, $Tau^2 = 0.17$). Single-factor analyses showed small differences in the magnitude of improvement between HT and VT in vertical jump ($g = 0.21$ [$-0.1, 0.52$], $p = 0.17$, $I^2 = 43.69$, $Tau^2 = 0.15$) and horizontal jump ($g = -0.15$ [$-0.53, 0.22$], $p = 0.40$, $I^2 = 51.60$, $Tau^2 = 0.20$) (Table 3).

For pooled short and medium distance sprint performance, moderate to large and significant improvements were evident with VT ($g = 0.64$ [$0.18, 1.1$], $p = 0.01$) and HT ($g = 1.04$ [$0.25, 1.82$], $p = 0.01$). High statistical heterogeneity was found (VT: $I^2 = 62.21$, $Tau^2 = 0.36$; HT: $I^2 = 81.41$, $Tau^2 = 1.09$). For the between-group meta-analysis, there was a small and non-significant effect in favor of HT ($g = -0.23$ [$-0.56, 0.10$], $p = 0.16$) (Fig. 4). Moderate statistical heterogeneity was found ($I^2 = 65.99$, $Tau^2 = 0.42$). Single-factor analyses showed small differences in the magnitude of improvement between HT and VT in short ($g = -0.33$ [$-0.85, 0.19$], $p = 0.19$, $I^2 = 71.30$, $Tau^2 = 0.55$) (Table 3.) and medium ($g = -0.12$ [$-0.37, 0.13$], $p = 0.28$, $I^2 = 0$, $Tau^2 = 0$) distance sprint (Table 3).

For pooled vertical and horizontal maximal strength, large increases were found with VT ($g = 0.88$ [$-0.20, 1.96$], $p = 0.09$) and HT ($g = 0.79$ [$0.12, 1.45$], $p = 0.03$). Moderate statistical heterogeneity was found (VT: $I^2 = 74.43$, $Tau^2 = 0.64$; HT: $I^2 = 65.39$, $Tau^2 = 0.46$). For the between-group meta-analysis, a small and non-significant effect was observed in favor of VT in improving maximal strength ($g = 0.27$ [$-0.56, 1.09$], $p = 0.42$) (Fig. 5A). High statistical heterogeneity was found ($I^2 = 84.37$, $Tau^2 = 1.25$). Single-factor analyses showed large differences in the magnitude of improvement between HT and VT in maximal strength tests conducted in horizontal ($g = -0.83$ [$-2.45, 0.79$], $p = 0.16$,

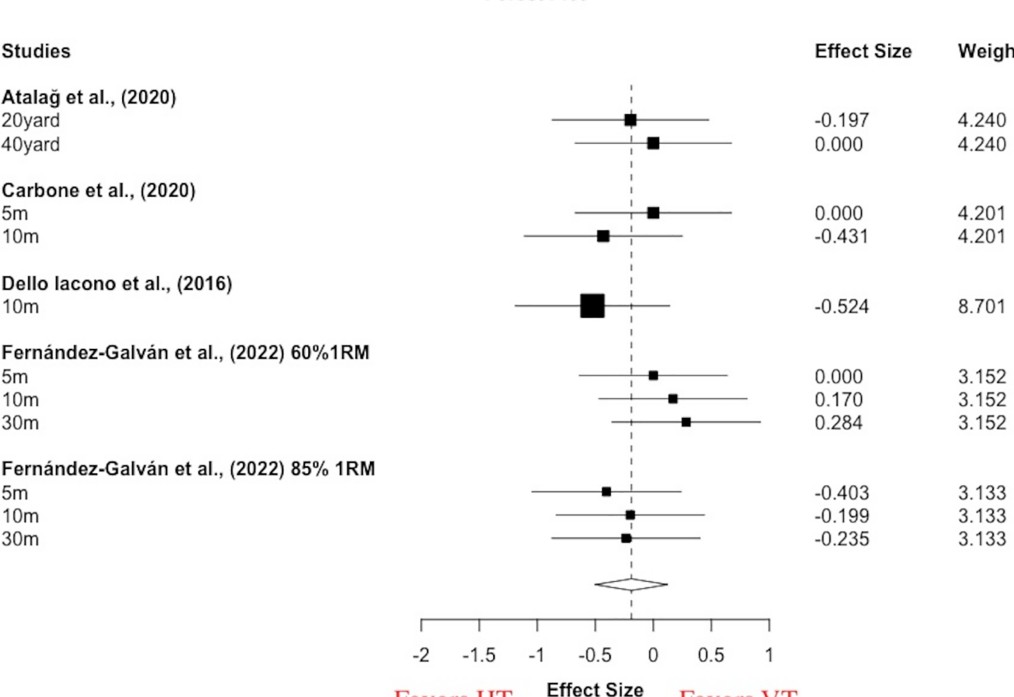

**Figure 2 Acute effects of vertical training *vs.* horizontal training on sprint performance.** Negative effect sizes denote in favoring of horizontal training for sprint performance (g = −0.19 [−0.51, 0.13], *p* = 0.17, I 2 = 0%, Tau 2 = 0). RM, repetition maximal; m, meter.

$I^2$ = 49.11, $Tau^2$ = 0.21) (Table 3) and vertical (g = 0.78 [−0.97, 2.53], *p* = 0.28, $I^2$ = 85.42, $Tau^2$ = 1.41) direction (Table 3).

For change of direction speed performance, significant moderate and larger improvements were observed with VT (g = 0.51 [0.16, 0.85], *p* = 0.01) and HT (g = 0.78 [0.11, 1.45], *p* = 0.03). Small to moderate statistical heterogeneity was found (VT: $I^2$ = 42.91, $Tau^2$ = 0.16; HT: $I^2$ = 70.65, $Tau^2$ = 0.58). For the between-group meta-analysis, a moderate and non-significant effect in favor of HT (g = −0.45 [−1.31, 0.41], *p* = 0.26) (Fig. 5B). Moderate statistical heterogeneity was found ($I^2$ = 74.57, $Tau^2$ = 0.66).

The results of subgroup analyses were presented in Table 3. The results showed no statistical significance in all subgroup analyses.

For within-group analyses, the results showed potential publication bias leading to potential overestimation of the effect of VT on jump (Z = 3.66, *p* = 0.0002), sprint (Z = 7.29, *p* < 0.001), and maximal strength (Z = 2.41, *p* = 0.02), and also potential overestimation of the effect of HT on jump (Z = 9.28, *p* < 0.0001), sprint (Z = 9.61, *p* < 0.001), maximal strength (Z = 2.15, *p* = 0.03), and COD and maneuverability (Z = 2.61, *p* = 0.009). For between-group analyses, potential underestimation of effect sizes was observed in sprint (Z = −3.09, *p* = 0.002) and COD and maneuverability (Z = −3.86, *p* = 0.001), and potential overestimation in maximal strength (Z = 2.15, *p* = 0.03).

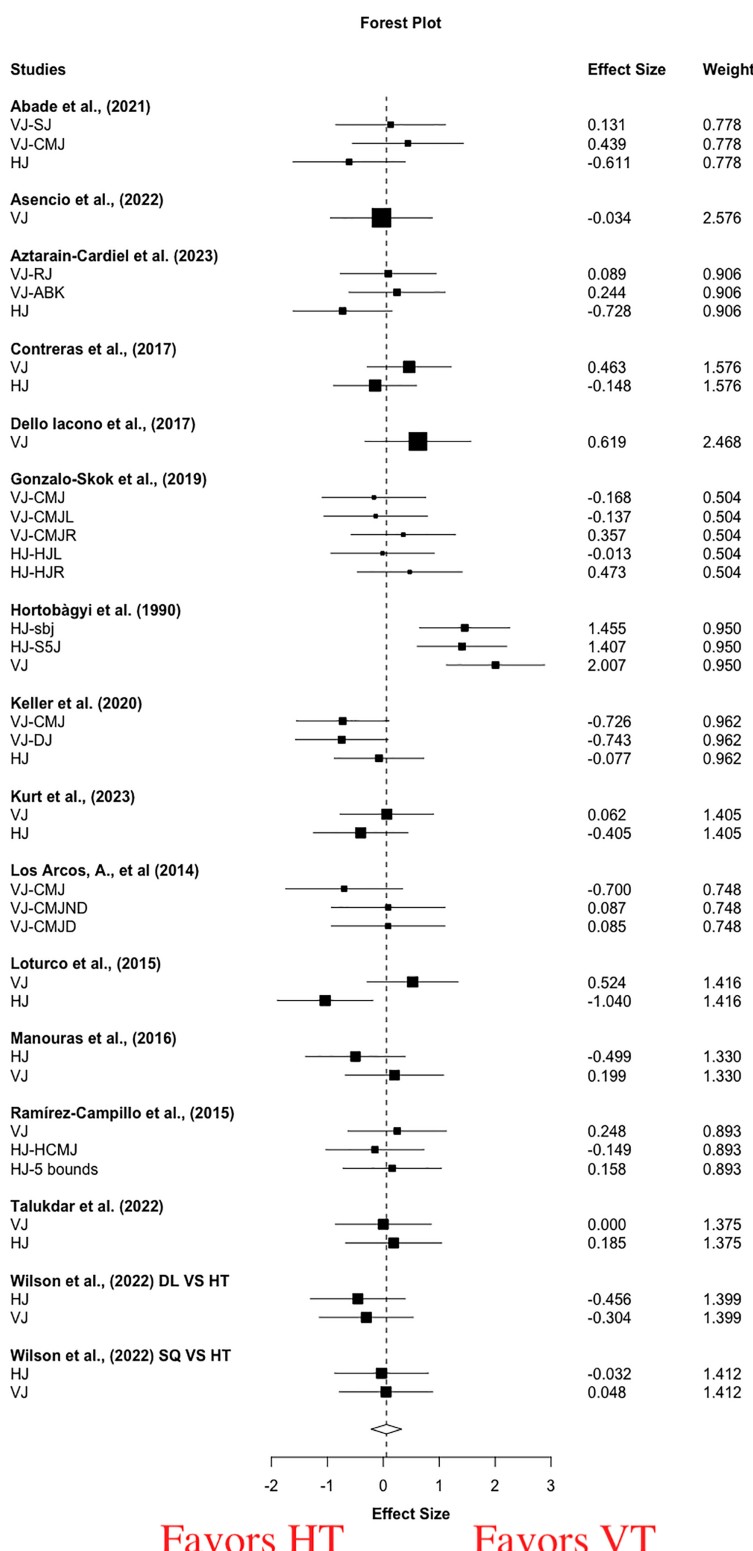

**Favors HT**          **Favors VT**

**Figure 3 Short-term effects of vertical training *vs*. horizontal training on jump performance.** Positive effect sizes denote in favoring of vertical training for jump performance (g = 0.06 [−0.22, 0.33], *p* = 0.67, I 2 = 46.46, Tau 2 = 0.17). VJ, vertical jump; HJ, horizontal jump; SJ, squat jump; CMJ, countermovement jump; RJ, rocket jump; ABK, abalakov jump; CMJL, leftleg countermovement jump; CMJR, right-leg

**Figure 3** (continued)
countermovement jump; HJL, left-leg hurdle jump; HJR, right-leg hurdle jump; DJ, drop jump; CMJND, non-dominant countermovement jump; CMJD, dominant countermovement jump; HCMJ, horizontal countermovement jump; sbj, standing broad jump; S5J, standing five jumps.

**Table 3 The results of subgroup analyses.**

| Performance | Moderators | | Effect size | | 95% CI | | | Heterogeneity | | Frequencies | |
|---|---|---|---|---|---|---|---|---|---|---|---|
| | | | g | SE | LL | UL | p | I$^2$ | Tau$^2$ | No | Ns |
| Sprint | Training modality | RT | −0.36 | 0.06 | −1.17 | 0.45 | 0.11 | 0.00 | 0.00 | 4.00 | 2.00 |
| | | PT | −0.22 | 0.21 | −0.71 | 0.27 | 0.32 | 74.69 | 0.65 | 26.00 | 9.00 |
| | Age | >18 yr | −0.30 | 0.34 | −1.18 | 0.58 | 0.42 | 83.06 | 1.21 | 18.00 | 6.00 |
| | | ≤18 yr | −0.20 | 0.11 | −0.48 | 0.09 | 0.13 | 0.00 | 0.00 | 14.00 | 6.00 |
| | Duration | >6 weeks | −0.46 | 0.29 | −1.20 | 0.28 | 0.17 | 73.97 | 0.72 | 14.00 | 6.00 |
| | | ≤6 weeks | −0.04 | 0.15 | −0.42 | 0.34 | 0.80 | 56.41 | 0.25 | 18.00 | 6.00 |
| | Sessions | >12 sessions | −0.56 | 0.35 | −1.54 | 0.42 | 0.19 | 78.36 | 0.98 | 12.00 | 5.00 |
| | | ≤12 sessions | −0.04 | 0.12 | −0.35 | 0.26 | 0.75 | 48.36 | 0.18 | 20.00 | 7.00 |
| | Single-factor | Short | −0.33 | 0.24 | −0.85 | 0.19 | 0.19 | 71.30 | 0.55 | 26.00 | 12.00 |
| | | Medium | −0.12 | 0.10 | −0.37 | 0.13 | 0.28 | 0.00 | 0.00 | 6.00 | 6.00 |
| Maximal strength | Training modality | RT | 0.35 | 0.37 | −0.84 | 1.54 | 0.42 | 88.10 | 1.82 | 9.00 | 4.00 |
| | Age | >18 yr | 0.68 | 0.28 | −0.55 | 1.90 | 0.14 | 87.93 | 2.09 | 5.00 | 3.00 |
| | | ≤18 yr | −0.30 | 0.27 | −3.71 | 3.11 | 0.46 | 71.80 | 0.50 | 6.00 | 2.00 |
| | Duration | >6 weeks | 0.61 | 0.64 | −7.53 | 8.74 | 0.52 | 93.95 | 4.47 | 4.00 | 2.00 |
| | | ≤6 weeks | 0.05 | 0.34 | −1.42 | 1.52 | 0.90 | 69.53 | 0.52 | 7.00 | 3.00 |
| | | ≤12 sessions | 0.35 | 0.37 | −0.84 | 1.54 | 0.42 | 88.10 | 1.82 | 9.00 | 4.00 |
| | Single-factor | Vertical | 0.78 | 0.63 | −0.97 | 2.53 | 0.28 | 85.42 | 1.41 | 8.00 | 5.00 |
| | | Horizontal | −0.83 | 3.72 | −2.45 | 0.79 | 0.16 | 49.11 | 0.21 | 3.00 | 3.00 |
| COD | Training modality | PT | −0.49 | 0.43 | −1.56 | 0.57 | 0.30 | 77.40 | 0.77 | 11.00 | 7.00 |
| | Age | >18 yr | −0.85 | 0.92 | −3.80 | 2.10 | 0.42 | 87.82 | 1.95 | 6.00 | 4.00 |
| | | ≤18 yr | −0.23 | 0.18 | −0.81 | 0.35 | 0.29 | 0.00 | 0.00 | 7.00 | 4.00 |
| | Duration | >6 weeks | −2.06 | 1.95 | −26.80 | 22.70 | 0.48 | 93.70 | 7.16 | 3.00 | 2.00 |
| | | ≤6 weeks | −0.11 | 0.17 | −0.54 | 0.32 | 0.54 | 10.88 | 0.02 | 10.00 | 6.00 |
| | Sessions | ≤12 sessions | −0.11 | 0.17 | −0.54 | 0.32 | 0.54 | 10.88 | 0.02 | 10.00 | 6.00 |
| Jump | Training modality | RT | −0.04 | 0.10 | −0.32 | 0.24 | 0.69 | 0 | 0 | 10.00 | 5.00 |
| | | PT | 0.10 | 0.18 | −0.30 | 0.51 | 0.58 | 60.30 | 0.31 | 29.00 | 11.00 |
| | Age | >18 yr | −0.07 | 0.1 | −0.31 | 0.16 | 0.48 | 15.03 | 0.04 | 16.00 | 8.00 |
| | | ≤18 yr | 0.17 | 0.23 | −0.37 | 0.71 | 0.47 | 61.58 | 0.31 | 23.00 | 8.00 |
| | Duration | >6 weeks | 0.35 | 0.30 | −0.41 | 1.12 | 0.29 | 65.47 | 0.42 | 14.00 | 6.00 |
| | | ≤6 weeks | −0.12 | 0.07 | −0.28 | 0.05 | 0.16 | 0 | 0 | 25.00 | 10.00 |
| | Sessions | >12 sessions | 0.33 | 0.30 | −0.45 | 1.1 | 0.33 | 72.31 | 0.57 | 14.00 | 6.00 |
| | | ≤12 sessions | −0.10 | 0.07 | −0.27 | 0.06 | 0.19 | 0 | 0 | 25.00 | 10.00 |
| | Single-factor | Vertical | 0.21 | 0.15 | −0.1 | 0.52 | 0.17 | 43.69 | 0.15 | 23.00 | 16.00 |
| | | Horizontal | −0.15 | 0.17 | −0.53 | 0.22 | 0.40 | 51.60 | 0.20 | 16.00 | 13.00 |

**Note:**
g, effect size; SE, standard error of the effect size; CI, LL, UL, 95% conûdence interval lower and upper limit; p, probability; I 2, percent of heterogeneity; Tau 2, absolute value of true heterogeneity; Ns, number of studies; No, number of outcomes; RT, resistance training; PT, plyometric training; COD, change of direction.

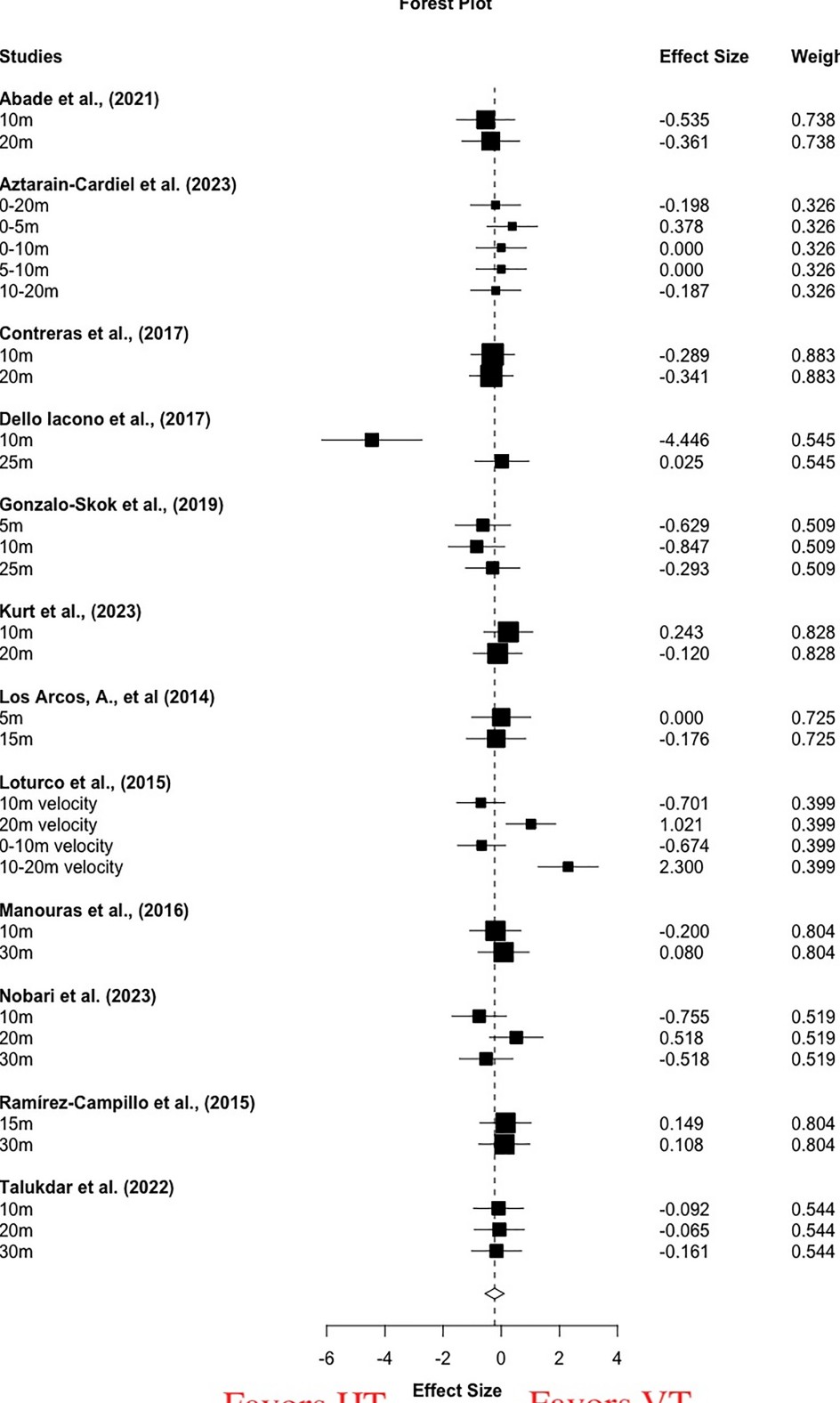

**Figure 4 Short-term effects of vertical training *vs* horizontal training on sprint performance.** Negative effect sizes denote in favoring horizontal training (g = −0.23 [−0.56, 0.10], *p* = 0.16, I 2 = 65.99, Tau 2 = 0.42).

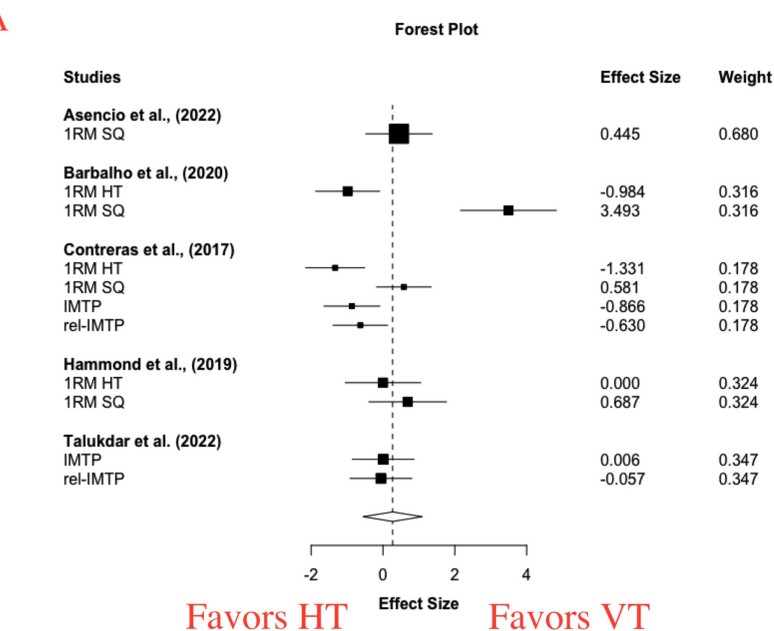

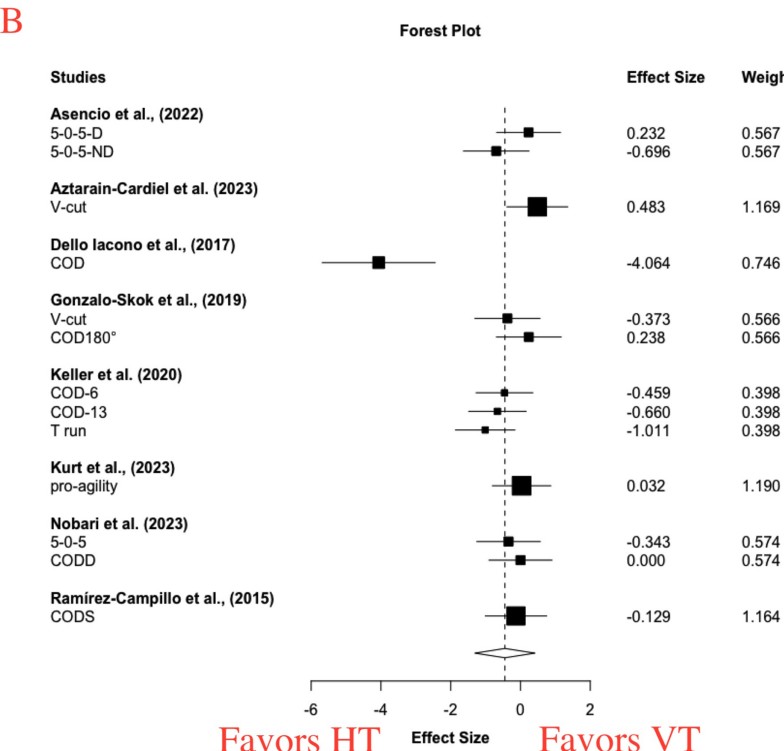

**Figure 5 Short-term effects of vertical training *vs*. horizontal training on maximal strength (A) and change of direction (B).** (A) Positive effect sizes denote favoring vertical training for maximal strength (g = 0.27 [−0.56, 1.09], *p* = 0.42, I 2 = 84.37, Tau 2 = 1.25). RM, repetition maximal; HT, hip thrust; SQ, squat; IMTP, isometric mid-thigh pull; rel-IMTP, relative maximal force of isometric midthigh pull. (B) Negative effect sizes denote favoring horizontal training for change of direction performance (g = −0.45 [−1.31, 0.41], *p* = 0.26, I 2 = 74.57, Tau 2 = 0.66). D, dominant-leg; ND, non-dominant leg; COD, change of direction; COD-6, 6 times COD; COD-13, 13 times COD; CODS, change of direction speed

## DISCUSSION

This systematic review aimed to compare direction specific resistance and plyometric training (HT *vs*. VT) on athletic performance, vertically and horizontally oriented performance outcomes. For acute studies, main findings from a limited number of studies suggested small and insignificant differences between HT and VT in sprint performance. Moreover, for short-term intervention, small to large but non-significant effect existed between HT and VT for improving athletic performance, vertically and horizontally oriented athletic performance outcomes, and short and medium distance sprint performance. Cumulatively, acute and short-term HT and VT were equally effective strategies for enhancing athletic performance, and vertically and horizontally oriented performance outcomes.

The acute effects of HT and VT on sprint performance were examined in four studies and 71 participants (*Atalağ et al., 2020*; *Carbone et al., 2020*; *Dello Iacono, Martone & Padulo, 2016*; *Fernández-Galván et al., 2022*) (Table 2). However, the meta-analysis was only conducted for sprint performance due to limited studies on jump (two outcomes from two studies) (*Carbone et al., 2020*; *Dello Iacono, Martone & Padulo, 2016*) and COD and maneuverability (one outcome from one study) (*Dello Iacono, Martone & Padulo, 2016*). The acute effects of a conditioning activity on subsequent physical performance were examined in the context of post-activation potentiation enhancement (*Seitz & Haff, 2016*). The form of activity to induce PAPE in the studies reviewed here comprised resistance training (squat *vs*. hip thrust) (*Atalağ et al., 2020*; *Carbone et al., 2020*; *Fernández-Galván et al., 2022*) and plyometric training (horizontal-alternate DJ *vs*. vertical-alternate DJ) (*Dello Iacono, Martone & Padulo, 2016*). Neither HT nor VT did improve sprint performance which was determined by within-group meta-analyses. Our findings showed non-significant changes in sprint performance following HT and VT. In contrast to the current results, a previous meta-analysis reported moderate effect sizes with respect to the pre-conditioning effects on sprint times (*Seitz & Haff, 2016*). The inconsistent finding might be largely explained by training loads and interval duration, but this is beyond the aim of the review.

In terms of sprint performance, the between-group meta-analysis revealed a small and non-significant effect in favor of HT (Fig. 2), which is expected in light of the non-significant within-group acute effects described above. The differences in the magnitude of PAPE responses might be influenced by several factors including the level of muscle strength, dose of the pre-conditioning activity, intervention modality, and other factors (*Seitz & Haff, 2016*). Notably, a single-group repeated-measures experimental design was conducted in all of these studies, implying that the results of between-group meta-analysis were not meaningfully affected by individual differences. The equivalent training dose was utilized for HT and VT, so that the training loads also did not meaningfully influence the results. A previous study reported that the PAPE magnitude is largely determined by the interaction of fatigue and potentiation (*Rassier & Macintosh, 2000*), suggesting that sprint performance will be improved if potentiation is greater than fatigue, and *vice versa*. In this case, there might be an optimal window to evoke a maximal

potentiating effect by a certain type of pre-conditioning activity, suggesting that the magnitude of PAPE is time dependent. Therefore, this finding seems to be unsurprising, and the effect of force direction executed by tasks is not significant.

Of the 15 included studies with 448 participants investigating the short-term effects of HT *vs.* VT on jump performance (*Abade et al., 2021*; *Asencio et al., 2022*; *Aztarain-Cardiel et al., 2023*; *Contreras et al., 2017*; *Dello Iacono et al., 2017*; *Gonzalo-Skok et al., 2019*; *Keller et al., 2020*; *Kurt et al., 2023*; *Los Arcos et al., 2014*; *Loturco et al., 2015*; *Manouras et al., 2016*; *Ramírez-Campillo et al., 2015*; *Talukdar et al., 2022*; *Tibor, Judit & Zoltán, 1990*; *Wilson et al., 2022*), intervention modalities of resistance training, plyometric training and a combination of both were conducted in four (*Abade et al., 2021*; *Asencio et al., 2022*; *Contreras et al., 2017*; *Wilson et al., 2022*), 10 (*Aztarain-Cardiel et al., 2023*; *Dello Iacono et al., 2017*; *Gonzalo-Skok et al., 2019*; *Keller et al., 2020*; *Kurt et al., 2023*; *Loturco et al., 2015*; *Manouras et al., 2016*; *Ramírez-Campillo et al., 2015*; *Talukdar et al., 2022*; *Tibor, Judit & Zoltán, 1990*), and one (*Los Arcos et al., 2014*) studies, respectively (Table 2). We examined the jump performance including jump height (16 studies, 23 outcomes, 448 participants) and distance (13 studies, 16 outcomes, 386 participants). Vertical jump tests involved bilateral CMJ (*Abade et al., 2021*; *Asencio et al., 2022*; *Contreras et al., 2017*; *Dello Iacono et al., 2017*; *Gonzalo-Skok et al., 2019*; *Keller et al., 2020*; *Kurt et al., 2023*; *Los Arcos et al., 2014*; *Loturco et al., 2015*; *Manouras et al., 2016*; *Ramírez-Campillo et al., 2015*; *Talukdar et al., 2022*; *Tibor, Judit & Zoltán, 1990*; *Wilson et al., 2022*), single-leg CMJ (*Los Arcos et al., 2014*), DJ (*Keller et al., 2020*), squat jump (*Abade et al., 2021*), rocket jump (*Aztarain-Cardiel et al., 2023*), and Abalakov jump (*Aztarain-Cardiel et al., 2023*). Horizontal jump tests included bilateral standing long jump (*Abade et al., 2021*; *Aztarain-Cardiel et al., 2023*; *Contreras et al., 2017*; *Keller et al., 2020*; *Kurt et al., 2023*; *Loturco et al., 2015*; *Manouras et al., 2016*; *Ramírez-Campillo et al., 2015*; *Talukdar et al., 2022*; *Tibor, Judit & Zoltán, 1990*; *Wilson et al., 2022*), single-leg standing long jump (*Gonzalo-Skok et al., 2019*), and horizontal five jumps (*Ramírez-Campillo et al., 2015*; *Tibor, Judit & Zoltán, 1990*). The result of the between-group meta-analysis revealed non-significant small effects in favor of VT for improving jump performance (Fig. 3), and non-significant small effect sizes favoring HT and VT for horizontal and vertical jump performance (Table 3), respectively, were observed in the single-factor meta-analyses.

The result of between-group meta-analysis was not surprising, as it could be explained by the fact that the jump performance including vertical and horizontal jump was analyzed. In this case, the contributions of VT and HT to the jump performance might be equivalent. These results of single-factor meta-analyses were consistent with that of a previous meta-analysis (*Junge, Jørgensen & Nybo, 2023*), the authors reported a small and non-significant effect size favoring VT in the vertical jump (SMD = −0.04, $p$ = 0.69), and a small and non-significant effect size favoring HT in horizontal jump (SMD = 0.25, $p$ = 0.07). These findings indicated no directional specificity of training on jump performance, thus refuting the force-vector theory again. Unexpected findings observed might be explained by several factors. From the mechanical perspective, the nature of the horizontal jump is multi-vectorial, suggesting the horizontal and vertical component forces are both critical for the horizontal jump performance. The horizontal jump

performance was determined by the vertical and horizontal velocities at take-off (*Hay, 1992*), both of which were the consequence of the vertical and horizontal decomposition of ground reaction forces (GRF), respectively. Evidence demonstrated that the relative peak values of vertical and horizontal ground reaction force accounted up 1.50–2.21% and 0.63–0.70% body mass, respectively, during a standing long jump task (*Wu et al., 2003*). However, the issue of horizontal forces is controversial from the mechanical reference perspective. The horizontal force is derived from the forward lean body position at the instant of take-off, and the horizontal force is actually the vertically based in the reference to the body frame (*i.e.*, local coordinate system) (*Fitzpatrick, Cimadoro & Cleather, 2019*). In this case, the distinction of body position at take-off is the primary difference between horizontal jump and vertical jump, yet the direction of force generated was aligned with the body orientation in both tasks. Although conflicting views existed in aforementioned perspectives, both can provide explanations for these unexpected findings.

Both vertical and horizontal jump involve lower-limb triple-extension, suggesting that the force is generated through rapid extension of the triple joints of the lower limbs during concentric phase (*i.e.*, proximal to distal sequencing). Thus, similar kinematic and kinetic characteristics of lower limb joints (*i.e.*, hip, knee, and ankle) might be expected in both tasks. Evidence revealed similarity in joint kinematics and small differences in joint moments and joint work of lower-limb joints during take-off between horizontal jump and vertical jump (*Fukashiro et al., 2005*). Discrepancy in joint moments and joint work is due to differences in muscle activity. For example, greater hamstring and lower rectus femoris activity was observed in horizontal jump, compared with vertical jump (*Fukashiro et al., 2005*). Differences in muscle activity have been attributed to a neural control strategy of optimizing jump performance, and the intrinsic muscle properties may not matter so much (*Fukashiro et al., 2005*). However, specific neural adaptations to direction-specific exercise may be minimal, as well-trained participants who were familiar with vertical and horizontal jump were analyzed in the review. Moreover, horizontal and vertical jump take-off velocity, and subsequent jump height/distance will be underpinned by lower-limb relative net impulsive capacity. Therefore, it is reasonable to assume that there is no significant difference in jumping performance between VT and HT.

In terms of the sprint performance, we identified 34 outcomes and 381 participants from 12 studies (*Abade et al., 2021*; *Aztarain-Cardiel et al., 2023*; *Contreras et al., 2017*; *Dello Iacono et al., 2017*; *Gonzalo-Skok et al., 2019*; *Kurt et al., 2023*; *Los Arcos et al., 2014*; *Loturco et al., 2015*; *Manouras et al., 2016*; *Nobari et al., 2023*; *Ramírez-Campillo et al., 2015*; *Talukdar et al., 2022*). Sprint time and velocity was examined in 11 (*Abade et al., 2021*; *Aztarain-Cardiel et al., 2023*; *Contreras et al., 2017*; *Dello Iacono et al., 2017*; *Gonzalo-Skok et al., 2019*; *Kurt et al., 2023*; *Los Arcos et al., 2014*; *Manouras et al., 2016*; *Nobari et al., 2023*; *Ramírez-Campillo et al., 2015*; *Talukdar et al., 2022*) and one studies (*Loturco et al., 2015*), respectively, and only two studies reported the split sprint performance (*i.e.*, 10–20 m) (*Aztarain-Cardiel et al., 2023*; *Loturco et al., 2015*) (Table 2). The sprint distance covered from 5 to 30 m, the short (≤20 m) and medium (20–40 m) distance sprint was examined by 12 studies (26 outcomes) (*Abade et al., 2021*; *Aztarain-Cardiel et al., 2023*; *Contreras et al., 2017*; *Dello Iacono et al., 2017*; *Gonzalo-Skok et al.,*

*2019*; *Kurt et al., 2023*; *Los Arcos et al., 2014*; *Manouras et al., 2016*; *Nobari et al., 2023*; *Ramírez-Campillo et al., 2015*; *Talukdar et al., 2022*) and six studies (six outcomes) (*Dello Iacono et al., 2017*; *Gonzalo-Skok et al., 2019*; *Manouras et al., 2016*; *Nobari et al., 2023*; *Ramírez-Campillo et al., 2015*; *Talukdar et al., 2022*), respectively. The between-group meta-analysis revealed non-significant small differences in the improvement magnitude of the sprint between HT and VT (Fig. 4), and single-factor analyses also showed similar enhancements in short- and medium-distance sprint between HT and VT. These findings suggested limited differences in enhancements in the overall, short, and medium sprint performance with HT and VT.

The force-vector theory suggests that the sprint task was performed in the anterior-posterior plane or horizontal direction, indicating that HT is potentially preferable for sprint performance, particular during initial acceleration. Unfortunately, our unexpected findings refuted this assumption. The biomechanical characteristics of sprint revealed that horizontal and vertical force components were critical determinants of sprint performance (*Mero, 1987*). During the sprinting acceleration phase, the horizontal and vertical force components accounted for 46% and 11% of the average forces, respectively (*Mero, 1987*), suggesting that horizontal forces play a more important role in sprinting acceleration. Thus, regardless of the development of muscle strength and power output, the improved vertical and horizontal effort following HT and VT may facilitate the development of sprint performance, albeit their contribution may be varied.

It is well documented that the higher the ratio of the horizontal force to total GRF (RF%) during the acceleration phase, the greater the forward speed (*Haugen, McGhie & Ettema, 2019*; *Morin, Edouard & Samozino, 2011*). However, for individual sprinters, the RF% is strongly determined by the sprint technique, instead of muscle quality (*Morin, Edouard & Samozino, 2011*). In other words, for an individual sprinter, the maximum horizontal force component will be reached when the athlete maintains their body leaning forward (*Haugen, McGhie & Ettema, 2019*) and a foot to ground impact that is behind or under their center of mass. The horizontal force component that is thus body and foot-to-ground impact orientation-dependent can only be improved from specific motor learning. Since the HT intervention shared greater similarities with the sprint task, it is theorized that HT could induce greater neural adaptations, such as learning effects and motor coordination, contributing to improved sprinting performance. Unfortunately, these positive adaptations may be trivial for well-trained participants who are familiar with sprint.

Moreover, the aforementioned horizontal force was in reference to the global frame, instead of the body frame. From the view of the body frame, the direction of force-generating (*i.e.*, resultant force) was in line with the body orientation during the sprint acceleration phase. The "horizontal force" is actually the vertically orientated *via* the body. In terms of the muscle quality, net forces over 100–300 ms (impulse) are essential elements of sprint (*Haugen, McGhie & Ettema, 2019*; *Suchomel, Nimphius & Stone, 2016*), though their contributions to sprint performance varied with the sprint distance increases. The development of muscle quality is dependent on the training loads, which were

performed similar with HT and VT, indicating that similar changes in sprint were thus logical.

However, *Junge, Jørgensen & Nybo (2023)* reported a moderate and significant effect size (SMD = 0.72, $p$ = 0.01) and a small and non-significant effect size (SMD = 0.03, $p$ = 0.83) favoring HT for the short- and long-distance sprint, respectively. Their findings were not in line with that of our review, but these inconsistent results can be explained by the following considerations. Firstly, the criteria for determining the short-distance sprint were varied. We classified the short- and medium- distance sprint as ≤20 and >20 and ≤40 m, respectively, whereas the short- and long-distance sprint were determined by the distance of ≤10 and >10 and ≤20 m, respectively, in a study by *Junge, Jørgensen & Nybo (2023)*. The characteristics of acceleration sprint is horizontal-force-dominant (*Morin et al., 2012*), while maximal speed sprint is vertical-force-dominant (*Nagahara et al., 2018*). The distance of sprint acceleration depends on the level of the athletes, most football and rugby athletes reached their maximal speed at the 15–20 m. Thus, the 0–20 m has been commonly recognized as the sprint acceleration phase. Secondly, the models of effect size estimation were varied. The 26 outcomes from 12 studies were utilized to analyze short-distance sprint in our review, thus the RVE method has to be conducted to avoid repeated weighting in our review.

The effects of HT and VT on change of direction speed performance were examined in 13 outcomes and 71 participants from 8 studies (Table 2) (*Asencio et al., 2022*; *Aztarain-Cardiel et al., 2023*; *Dello Iacono et al., 2017*; *Gonzalo-Skok et al., 2019*; *Keller et al., 2020*; *Kurt et al., 2023*; *Nobari et al., 2023*; *Ramírez-Campillo et al., 2015*). Between-group meta-analysis showed non-significant differences between HT and VT in the improvement magnitude of COD speed and maneuverability performance (Fig. 5B), and all subgroup analyses showed no statistical significances. This result therefore suggested HT as effective as VT in improving COD speed and maneuverability performance. This finding was in line with that of a previous review (*Junge, Jørgensen & Nybo, 2023*), which reported a small effect size favoring HT for improving COD performance, non-significance but a tendency of significance was observed (SMD = 0.31, $p$ = 0.06). However, 11 outcomes from seven studies were analyzed within one meta-analysis in their review, arising the concerns of repetitive weighting that would overestimate or underestimate the effect size.

These results were not surprising as the COD and maneuverability task was also multi-vectorial. Existing evidence showed vertical and horizontal propulsive force reached 14.8–15.15 and 11.39–11.69 N/kg, respectively, during 505 task (*Dos'Santos et al., 2017*). Although vertical and horizontal propulsive force have varying effects on COD performance, horizontal propulsion may have a greater contribution to 180° turning performance (*Dos'Santos et al., 2017*). However, we cannot ignore the positive effect of vertical propulsive force on the COD performance which contributes to overall resultant force generation which facilitates net acceleration. *Spiteri et al. (2015)* divided the participants into the faster and slower groups according to their COD performance, and found that the faster group generated greater vertical force compared to that of the slower group. These findings suggested that both vertical and horizontal forces are significant to the COD performance.

Furthermore, the COD performance was determined by multifactor. The 57% COD performance can be explained by sprint performance and muscle strength (*Young, Miller & Talpey, 2015*). For example, the COD performance was examined including the 505 test, V-cut test, T-run, and pro-agility in this review, and most of these COD and maneuverability tasks were strongly associated with sprint time (*Hernández-Davó et al., 2021*; *Pereira et al., 2018*). This strong relationship may be explained by the fact that the COD and maneuverability tasks were largely occupied by the sprint task. However, we found that both qualities were equally enhanced following HT and VT in the present review. Therefore, limited differences in improved COD and maneuverability were unsurprising between HT and VT.

The short-term effects of HT and VT on maximal strength were based on data from 123 participants and 11 effect sizes from the five studies (Table 2) (*Asencio et al., 2022*; *Barbalho et al., 2020*; *Contreras et al., 2017*; *Hammond et al., 2019*), the comparisons of horizontally- and vertically-oriented training exercises involved squat *vs.* hip thrust (*Barbalho et al., 2020*; *Contreras et al., 2017*; *Hammond et al., 2019*), and horizontal flywheel training *vs.* vertical flywheel training (*Asencio et al., 2022*). Interpretation of within-group meta-analyses results indicated that both HV and VT largely and significantly improved muscle strength, whereas HT was as effective as VT for muscle strength gains and maximal strength tests conducted in horizontal and vertical directions, as indicated by small to large, non-significant effect sizes from the findings of between-group and single-factor meta-analyses. Moreover, the result of between-group meta-analysis was not meaningfully influenced by all analyzed moderators including age, intervention modality, duration, frequency, and total sessions (Table 3).

Notably, although two previous meta-analyses shared a similar topic with the current review (*Junge, Jørgensen & Nybo, 2023*; *Moran et al., 2021*), these reviews did not analyze the effects of HT and VT on maximal strength, and maximal strength tests conducted in both directions. In the present meta-analysis, the maximal strength performance was examined including vertically (eight outcomes five studies) and horizontally (three outcomes three studies) oriented maximal strength tests, it thus is reasonable to assume the similarity in muscle strength gains with HT and VT.

Surprisingly, we did not find directional specificity of training on maximal strength. These findings supported our hypothesis, whereas refuted the force-vector theory. Firstly, the development of muscle strength was largely determined by the training loads (*Lopez et al., 2021*), instead of the direction of force executed during training exercises. All of five included studies reported that equivalent training loads were utilized for HT and VT, similar muscle strength gains might thus be expected. Secondly, the key element of resistance training was to develop the targeted muscle. These vertically- and horizontally-oriented lower-limb maximal strength tasks both required triple extension of lower-limb joints (hip, knee, and ankle), suggesting similar force-generating muscle groups were involved. For example, squat (vertical task) and hip thrust (horizontal task) simultaneously and similarly activated gluteus maximus, biceps femoris, *etc.* (*Contreras et al., 2015*; *Delgado et al., 2019*). Notably, three of five included studies compared the short-term effects of squat *vs.* hip thrust in this review (*Barbalho et al., 2020*; *Contreras*

_et al., 2017_; _Hammond et al., 2019_). In this aspect, limited differences in muscle activation might be observed between HT and VT, and the target muscles for maximal strength testing in the horizontal and vertical directions were also similar. Thus, non-significant improvements in maximal strength gains were expected following HT and VT. Thirdly, for well-trained players, morphology adaptions may be greater than neural adaptations following both interventions in the initial duration. The muscle strength gains resulted from morphology and neural adaptions to resistance training and neural adaptions commonly play a dominant role in initial strength adaptions (durch Krafttraining 2007). Notably, the well-trained participants were analyzed in those five studies, of which three studies reported that participants had a minimum of 6 months to 3 years of resistance training experience (_Barbalho et al., 2020_; _Contreras et al., 2017_; _Hammond et al., 2019_), and two studies reported that participants had 1–12 years experience in their respective sports (_Asencio et al., 2022_; _Talukdar et al., 2022_). Thus, there were fewer neural adaptions to initial strength adaptions, suggesting that the learning effect may be negligible.

The limited number of studies regarding acute studies (four studies) and maximal strength (five studies) suggested more studies are required to be conducted in the future with respect to directional training. Future studies need to consider the effect of training experience on participants. We only analyzed the recreational athletes, where morphology adaptation had a greater effect on their muscle strength gains. However, for novel players, the neural adaptions may play a crucial role in the initial muscle strength gains, suggesting the learning effect may influence the directional-specific adaptions. In this regard, athletic performance outcomes would benefit from exercise training that is more similar to performance outcomes. For example, the hip thrust may be superior to the back squat in improving hip thrust 1RM in the novel players. Moreover, moderate to high heterogeneity was observed for short sprints, maximal strength and COD, which may be due to differences in participants and exercise regimens, further weakening the ability to derive evidence-based recommendations.

## Practical applications

The theory of force-vector specificity of training exercises involves force-vector and dynamic correspondence theory and has been communicated previously (_Fitzpatrick, Cimadoro & Cleather, 2019_; _Goodwin & Cleather, 2016_). However, the results of this systematic review and meta-analysis suggest that this is not reflected in the current empirical evidence, indicating that comparable improvements in sprint, COD, and horizontal and vertical jump were observed following HT and VT. Therefore, the force-vector characteristics of a specific sport can be disregarded in the program design, and the inclusion of vertical (_e.g._, squat and drop jump) and horizontal (_e.g._, hip trust and broad jump) force-vector exercises in the program design may enhance the training variety. However, Practitioners should consider the relevance of the selected movement characteristics as well as the specific goals of the training. For example, barbell hip thrust training is preferred for athletes aiming to improve gluteus maximus strength because the barbell hip thrust activates the gluteus maximus more than the back squat (_Contreras et al., 2015_).

## CONCLUSIONS

Overall, the current meta-analysis showed that HT and VT are equally effective for enhancing athletic performance outcomes, vertically and horizontally oriented performance outcomes. These findings refuted the directional specificity of training on performance outcomes.

### Funding

The authors received no funding for this work.

### Competing Interests

The authors declare that they have no competing interests.

### Author Contributions

- Jiaru Huang conceived and designed the experiments, performed the experiments, analyzed the data, prepared figures and/or tables, authored or reviewed drafts of the article, and approved the final draft.
- Tibor Hortobágyi conceived and designed the experiments, performed the experiments, analyzed the data, prepared figures and/or tables, authored or reviewed drafts of the article, and approved the final draft.
- Thomas Dos'Santos conceived and designed the experiments, performed the experiments, analyzed the data, prepared figures and/or tables, authored or reviewed drafts of the article, and approved the final draft.
- Yu Shi conceived and designed the experiments, performed the experiments, analyzed the data, prepared figures and/or tables, authored or reviewed drafts of the article, and approved the final draft.
- Yilin Que conceived and designed the experiments, performed the experiments, analyzed the data, prepared figures and/or tables, authored or reviewed drafts of the article, and approved the final draft.
- Junlei Lin conceived and designed the experiments, performed the experiments, analyzed the data, prepared figures and/or tables, authored or reviewed drafts of the article, and approved the final draft.
- Yuying Su conceived and designed the experiments, performed the experiments, analyzed the data, prepared figures and/or tables, authored or reviewed drafts of the article, and approved the final draft.
- Wei Li conceived and designed the experiments, performed the experiments, analyzed the data, prepared figures and/or tables, authored or reviewed drafts of the article, and approved the final draft.

### Data Availability

This is a systematic review/meta-analysis.

## Supplemental Information

Supplemental information for this article can be found online at http://dx.doi.org/10.7717/peerj.18047#supplemental-information.

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
