# Peer review of "Effects of direction specific exercise training on athletic performance: a systematic review and meta-analysis"

_PeerJ, doi:10.7717/peerj.18047_

## Round 0.1 · original submission · Major Revisions

Dear Authors:

After excellent, in my opinion, peer-reviews, I consider "Major Reviews".

Please attend to the reviewer´s comments.

Regards

Dr. Manuel Jiménez

Reviewer 1 ·

Basic reporting

The primary purpose of the meta-analysis was to compare HT vs VT on vertical and horizontal performance outcomes. Outcomes included horizontal and vertical jumps, sprints, change of direction…

The authors have addressed a long-debated topic of specificity of training in whether to focus on vertical or horizontal training or force-vector vs dynamic correspondence.
There are some issues with the manuscript that need to be addressed.
The authors use numerous review articles and books as references. Primary references should be utilized. Many of the references are incomplete with the volume and pages missing. The authors are also inconsistent in the names of the journals. Some journal names are abbreviated, and some are written out. The reference by Prins et al. (Line 164) does not include a date and is not appropriate for this article. The Prins et al reference (line 802) is titled, “Investigating the Causal Relationship of C-Reactive Protein with 32 Complex Somatic and Psychiatric Outcomes: A Large-Scale Cross-Consortium Mendelian Randomization Study”

Experimental design

There are no issues with the experimental design.
The authors followed the suggests of Basu (2017, https://peerj.com/preprints/2978v1/) and provided the PRISMA 2020 checklist.

Validity of the findings

Standard heterogeneity ranges include: 0% to 40%: might not be important; 30% to 60%: may represent moderate heterogeneity; 50% to 90%: may represent substantial heterogeneity; and 75% to 100%: considerable heterogeneity. Based on Table 3, 63% of the results (22 out of 35 variables) represent substantial or considerable heterogeneity. The large amount of heterogeneity is really not addressed in the article.

The ability to determine statistical differences in performance variables do not always match athletic performance outcomes, e.g. five seconds between first and second may not be statistically significantly different but is different from a performance outcome. Throughout the results section, the authors state small and non-statistically significant changes. There is no change or no difference if it is non-statistically significant.

Line 369-370 – Awkward sentence.
Line 376-377 - If non-significant, no difference and does not favor either HT or VT.

Non-significant is used frequently, followed by ... favoring HT and VT. If it is non-significant, then there is no favorable difference.

Line 487 – 488 - This statement is incorrect. The Mero et al. article states...Vertical peak-to-peak displacement of the center of gravity of the body during the stride has been shown to decrease with increased running speed. Page 382, right column. On Page 384 of same article, section 2.3.3 States – During constant velocity sprint running, increases in horizontal and vertical force production with increasing running speed have been reported (Mero & Komi 1986). Also, the investigators are referencing a review article when the original reference should be utilized.
Line 523 – 524 - The authors are referencing a review article in a trade journal. The original research should be utilized

Line 545 – Existing evidence…
Line 548 - … may have a greater…

Line 624 – the word “vertically” has different sized font within the word.

Additional comments

Check titles of journals. Some are abbreviated some are complete
Check references for completeness. For example Behm (1995) does not include page numbers.
durch Krafttraining NA reference. Check completeness.
Look for the others with similar issues.
Prins et al. Wrong reference

Table 1. Listing the studies in alphabetical order would be helpful for the reader to locate a specific paper.

Figures. While the results indicate there is really no difference between HT and VT, it would be helpful to identify on the figures which direction on the x-axis favors HT and VT. i.e., the left of the SMD favors HT and the right of SMD favors VT. It is included in the figure caption but labeling it on the graph would be helpful.

·

Basic reporting

Some terms and phrases may be complex for readers. Using simpler language could benefit those new to the topic.
More emphasis could be placed on acute intervention effects in the introduction, which could broaden the scope of the study.
A deeper exploration of conflicting findings in the existing literature would strengthen the motivation for the research.
Recommendations
Theoretical concepts could be reinforced with examples. For instance, brief examples of the successes of different training types could be included.
Establishing relationships between different aspects of training could give the introduction a more integrated view.
Adding a brief summary at the end of the introduction outlining how the study will contribute to the field could engage the reader’s interest.

The introduction section lays a solid foundation by addressing important aspects of the topic. However, making the language more accessible and providing a deeper critical perspective on existing literature could strengthen the section. Overall, the paper presents an intriguing research objective, and with certain improvements, it could become even more effective.

Experimental design

The absence of a pre-registered review protocol may introduce potential biases. Future studies should consider registration to enhance transparency.
More detailed information about participant characteristics (e.g., age, gender distribution) would be beneficial for assessing the generalizability of the findings.
The data extraction process conducted by only two researchers may increase the risk of errors. Involving additional researchers could improve reliability.
Providing more details on how each step of the methodology was carried out would enhance reader understanding.
While the use of a quality assessment tool is a positive step, more information about this tool would provide greater transparency for the reader.
The methods section indicates that the study is built on a solid foundation. However, adding more detail and explanation in certain areas could improve the quality of the section. Overall, the methodology presents a good framework for this systematic review and meta-analysis.

Validity of the findings

The results section is generally well-structured and informative. With some improvements in clarity and depth of discussion, it can effectively convey the significance of the findings. Overall, it lays a solid foundation for interpreting the implications of the meta-analysis in the context of athletic training.

Additional comments

The article addresses a current issue and has the potential to make significant contributions to the literature; the research aim and methodology are clearly stated, and the findings are presented effectively. However, providing more details on the applicability of the methods and explicitly highlighting the limitations would be beneficial. The discussion section presents a strong comparison of the findings with the literature, but including practical recommendations could enhance the value of the results. While the language is fluent and scientific terminology is used correctly, some sections could benefit from a more straightforward expression. Overall, the article is well-structured and can be considered a successful study with areas for further improvement.

Reviewer 3 ·

Basic reporting

This paper is written and well-organized. The introduction and background are reasonable
given the premise of the paper. The figures and tables are comprehensive and helpful.

Experimental design

In general, the experimental design was excellent and written.

Validity of the findings

The results are reasonable given the experiments

Additional comments

This interesting paper addresses a need for the scientific.

---

## Round 0.2 · Minor Revisions

Dear Authors:

Thank you for waiting. Please attend to the remaining minor reviews.

Regards

Dr. Manuel Jiménez

Reviewer 1 ·

Basic reporting

Acceptable

Experimental design

Acceptable

Validity of the findings

Acceptable

Additional comments

The authors have updated the manuscript in response to most of my comments and will accept their responses for the items not updated.

Items not addressed and need correcting include the updating of references. Some references have abbreviated journals and others have the full name of the journal written out. As an example, the first reference (Abade et al) the title of the journal is complete with full words, no abbreviations. The second reference (Asadi et al.) the title of the journal uses the abbreviations of the words. This variability in journal titles occurs throughout the list of references. The authors need to follow PeerJ format for journal titles in the references.

Line 810: The original version of the manuscript included a reference for Mero (1987)
Mero A. 1987. Electromyographic activity, force and anaerobic energy production in sprint running: with special reference to different constant speeds ranging from submaximal to supramaximal.
It was requested that authors provide a complete reference. The authors provided the following.
Mero A. 1987. Electromyographic activity, force and anaerobic energy production in sprint running: with special reference to different constant speeds ranging from submaximal to supramaximal. Europ. J. Appl. Physiol 55, 553-561. 10.1007/BF00421652.
The journal reference and DOI is incorrect. The original reference is a dissertation and can be found here.
https://jyx.jyu.fi/bitstream/handle/123456789/71990/Mero_Antti_screen.pdf?sequence=4&isAllowed=y
The DOI from the second version of the paper is a completely different article.
https://link.springer.com/article/10.1007/BF00421652
This is unacceptable.

The entire reference list is assembled poorly. While the authors may have utilized Endnote or Mendeley, they clearly did not review the output to ensure correct formatting.

Line 731 Bmj is not correct.
Line 817 and Line 820. The two references by Morin et al. should be corrected. Morin JB. Remove the hyphen in the 817 reference.
Line 891: The DOI for this reference is https://doi.org/10.4015/S1016237203000286
Line 893: Verkhoshasky: This is a self-published book. So I am not quite sure how to reference it, as a book or as a website. You will need to check with the PeerJ to determine the best way. But I do not think the current format is correct.
Line 897: Zweifel and Michael. The authors name should be Zweifel MB.

DOI’s are available for the references that do not have them listed. Add DOI’s.

***Take the time to review all references for correct formatting. The items I listed above are a partial list.


Other items:
Line 173. The parentheses around (1) are in the incorrect place. It should read (1-repeition maximal,…)
Line 174: Add COD to the change of direction (COD). Add COD to the Keywords as well.
Line 296: correct reference [49]. Follow correct format for referencing.
Page 50 of 64. Figure 4 contains two x-axis’ with different scales. The figure title does not include information about what the two x-axis represent.

·

Basic reporting

necessary changes have been made. suitable

Experimental design

necessary changes have been made. suitable

Validity of the findings

necessary changes have been made. suitable

---

## Round 0.3 · accepted · Accept

Dear Authors:

Thank you for your patience and for improving your manuscript, "Effects of direction-specific exercise training on athletic performance: a systematic review and meta-analysis." We are accepting your manuscript for publication. Thank you for trusting PeerJ.

Congratulations

Dr. Manuel Jiménez

Reviewer 1 ·

Basic reporting

Acceptable

Experimental design

Acceptable

Validity of the findings

Acceptable.